# Amino acid substitutions in human growth hormone affect secondary structure and receptor binding

**Andrei Rajkovic[1], Sandesh Kanchugal[1], Eldar Abdurakhmanov[2], Rebecca Howard[3], Sebastian Wärmländer[3], Joseph Erwin [3], Hugo A. Barrera Saldaña[4], Astrid Gräslund[3], Helena Danielson [2], Samuel Coulbourn Flores [3,5]\***

**1** Department of Cell and Molecular Biology, Uppsala University, Uppsala, Sweden, **2** Department of Chemistry, Uppsala University, Uppsala, Sweden, **3** Department of Biochemistry and Biophysics, Stockholm University, Frescati, Sweden, **4** Department of Biochemistry, Autonomous University of Nuevo León, Monterrey, Mexico, **5** Department of Animal Breeding and Genetics, Swedish University of Agricultural Sciences, Uppsala, Sweden

\* samuel.flores@slu.se

**Data Availability Statement:** All relevant data are within the paper and its Supporting Information files.

## Abstract

The interaction between human Growth Hormone (hGH) and hGH Receptor (hGHR) has basic relevance to cancer and growth disorders, and hGH is the scaffold for Pegvisomant, an anti-acromegaly therapeutic. For the latter reason, hGH has been extensively engineered by early workers to improve binding and other properties. We are particularly interested in E174 which belongs to the hGH zinc-binding triad; the substitution E174A is known to significantly increase binding, but to now no explanation has been offered. We generated this and several computationally-selected single-residue substitutions at the hGHR-binding site of hGH. We find that, while many successfully slow down dissociation of the hGH-hGHR complex once bound, they also slow down the association of hGH to hGHR. The E174A substitution induces a change in the Circular Dichroism spectrum that suggests the appearance of coiled-coiling. Here we show that E174A increases affinity of hGH against hGHR because the off-rate is slowed down more than the on-rate. For E174Y (and certain mutations at other sites) the slowdown in on-rate was greater than that of the off-rate, leading to decreased affinity. The results point to a link between structure, zinc binding, and hGHR-binding affinity in hGH.

## Introduction

Human Growth Hormone (hGH) binds a single hGH Receptor (hGHR) using its Site 1, a large, physicochemically diverse binding region. It then recruits a second hGHR to bind at its lower-affinity Site 2 [1]. The hGHR dimerization initiates signaling through the JAK/STAT pathway. Thus one strategy to disrupt signaling is to prevent dimerization. This in turn can be done by destroying binding at site 2, which is easily effected with mutations such as G120K [2]. It is also useful to simultaneously strengthen binding at site 1 [3], and both were done in

**Funding:** We gratefully acknowledge support from the Swedish Foundation for International Cooperation in Research and Higher Education (STINT, IG2012-5157) to SF. The Lars Hierta Memorial Foundation provided salary support for AR. SF receives partial salary support from the Swedish Research Council grant VR-M 2016-06301, the National Research School in Medical Bioinformatics. The funders had no role in study design, data collection and analysis, decision to publish, or preparation of the manuscript.

**Competing interests:** The authors have declared that no competing interests exist.

Pegvisomant development. As part of that process the interesting substitution E174 was discovered which increases binding but whose mechanism could not be explained by Cunningham & Wells [4]. In this work we measure the kinetics and secondary-structural effects of that mutation, along with that of a different substitution at the same position (E174Y), a negative control (L52F, which is positioned far from position 174, and outside the helices), and the Wild Type (WT, the neutral control).

Pegvisomant is a recombinant hGH (rhGH) which was affinity matured at site 1 using phage display, reaching an affinity 400-fold higher than WT [3]. However, this generated substitutions at 15 amino acid positions in site 1, and other considerations required manually selected reversions and mutagenesis. The G120K mutation was also generated. Lastly, the rhGH was PEGylated to extend serum half-life. These manipulations resulted in significant reduction of affinity compared to WT [5]. As a result, there remains significant potential for rhGH variants to be used as therapeutics and diagnostics for several cancers and growth disorders, hence motivation to understand the biophysics of binding at site 1. We are interested in the possibility of choosing amino acid substitutions at a small number of residue positions, obtaining higher affinity while also working within constraints such as maintaining solubility and enabling modifications such as PEGylation. One could in principle also avoid patented substitutions, though that is not a concern in the case of Pegvisomant.

Cunningham & Wells discovered E174A during alanine scanning—a technique used to map binding interfaces in the absence of structural data. Alanine is smaller than all other canonical amino acids except glycine. Therefore protein-protein interactions (PPIs) are usually weakened if the substitution is at the interface. However E174A is an exception–it increases affinity [4]. E174, along with H18 and H21, is part of a zinc-binding triad [6] (Fig 1). E174 is also packed between two helices and follows the typical heptad repeat [7]. We therefore paid particular attention to E174. How could an amino acid which is part of the zinc binding triad, be mutated (to alanine, no less) and increase affinity?

More broadly, we are also interested in the general problem of computationally engineering proteins by introducing single-residue-position substitutions, especially to improve binding affinity [8]. Other computational-experimental methods have yielded considerable improvements in affinity, but only because they included an experimental high-throughput screening and affinity maturation stage [9,10]. The required expenditure is out of reach of many labs. The end product typically has substitutions at multiple positions, as occurred in [3], making it difficult to know which substitutions were most important and why. This lack of guiding knowledge is the likely reason some modifications of [5] were detrimental to binding. If one could start with a given scaffold, generate a small number of substitutions at single positions, and obtain improvements in affinity for even a small number of these, this would mean a better ability to 1) fine-tune protein properties without immunological, off-target and other unintended effects, 2) avoid patent-protected modifications, and 3) understand the evolutionary purpose of the native scaffold sequence. Thus in addition to E174A, E174Y, L52F, and WT, we also computationally selected 21 other substitutions at several single-amino-acid positions, including position 174, expressed the corresponding rhGH variants, and measured their effect on rhGH-hGHR binding. This screen did not result in any affinity-increasing substitutions other than E174A, but the results have interesting consequences for protein engineering in general, so they are included here as a secondary outcome of the work.

## Method

We used HomologyScanner [8] with FoldX [11] to identify single-residue substitutions in rhGH which have potential to increase affinity to hGHR. We expressed and purified a number

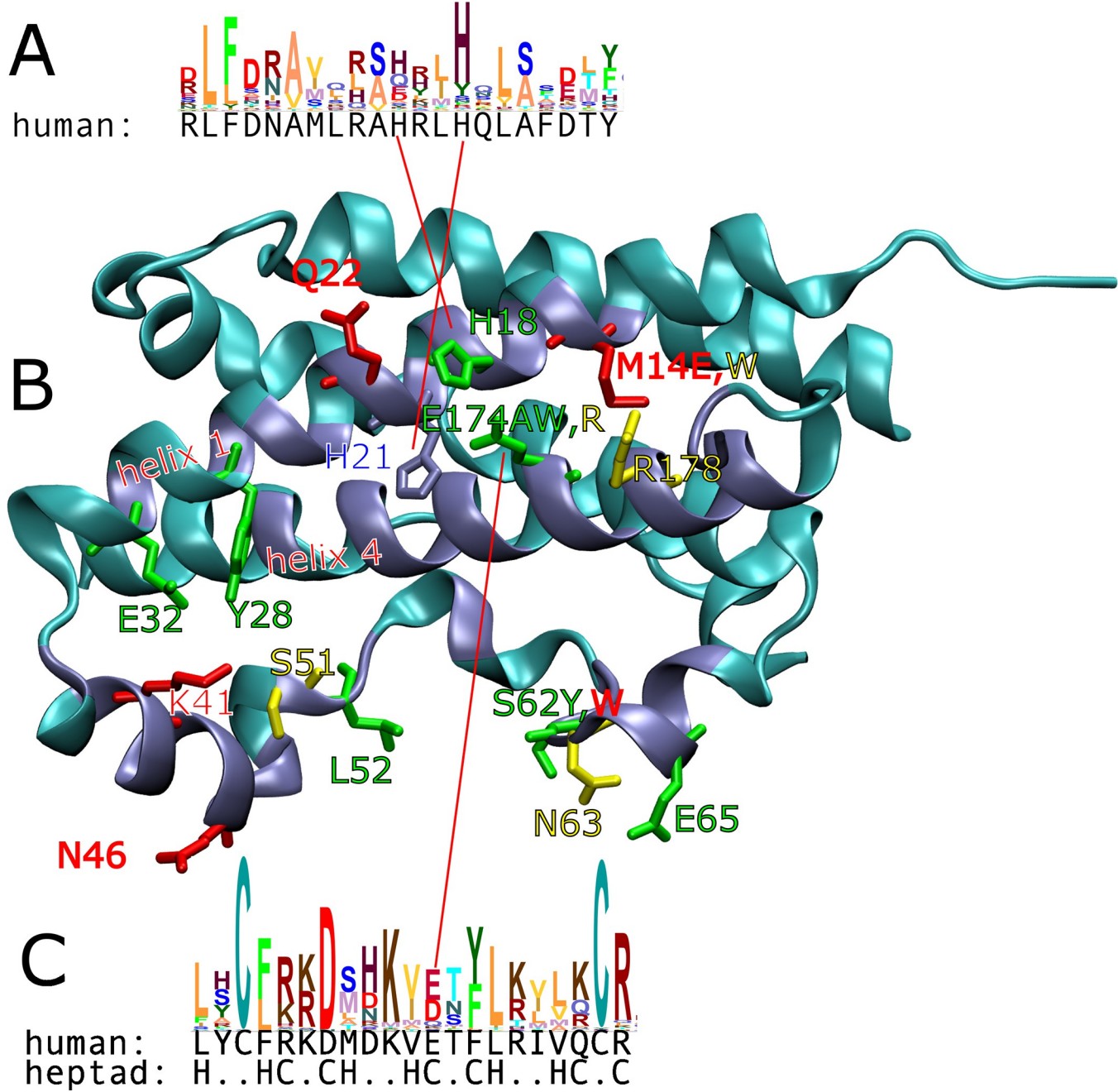

**Fig 1. Conserved position E174 is part of the zinc binding triad and obeys the heptad pattern.** Coiled-coils tend to follow the heptad pattern of HxxHCxC (H = hydrophobic, C = charged). (A) Helix 1 appears coiled with its antiparrallel partner helix 4. However for the former no coiled-coil signal was found by COILS, MultiCoil, or other expasy coiled-coil predictors. H18 and H21 are conserved (multiple sequence alignment from PFAM family PF00103, full alignment, profile generated with Skylign, vertical size proportional to information) and part of the zinc binding triad. Thus the triad appears important for coiled-coiling (B) in the hGH monomer. Site 1 residues are shown in iceblue, remaining residues in cyan cartoons. At several positions, we generated mutations with binding kinetics similar to those of WT (green labels). We identified a second cluster of mutations with similar koff, but much slower kon (red labels). Mutants with kinetics that did not fall into either cluster are labelled in yellow. At some positions different substitutions were tested which produced different kinetics (M14E,W, S62Y,W, E174A,W,R). (C) Helix 4 follows the heptad pattern closely, and is predicted to be in a coiled-coil by COILS [7]. E174 is conserved and part of the zinc binding triad.

of such variants, and measured their hGHR binding kinetics by Surface Plasmon Resonance (SPR). Lastly, we measured the Circular Dichroism (CD) spectra for WT, E174A, E174Y, and L52F.

## Computational selection of mutations

HomologyScanner is an automated service which computes the FoldX energy for a specified mutation using not one but several PDB structures (here 1A22, 1HWG, 1HWH, and 1AXI); [8] The average over the structures is reported as $\Delta\Delta G_{homologyScanner}$. We performed computational saturation mutagenesis over site 1. That is to say, we computed the effect of all 19 possible substitutions, at every position in site 1. Then we sorted by ascending $\Delta\Delta G_{homologyScanner}$. No more than one substitution of each physicochemical class (positive, negative, hydrophic, polar) was included in this list. We then selected the 30 mutants with lowest $\Delta\Delta G_{homologyScanner}$, for experimental evaluation. The HomologyScanner results are given in Table 1.

## Expression of hGH variants in E. coli

The selected rhGH variant DNA was generated by gene synthesis and cloned into pET-28b vectors by Genscript. The sequence included a hexahistidine tag at the N-terminus, in order to facilitate purification using a nickel column. Our "WT" rhGH sequence was thus:

HHHHHHFPTIPLSRLFDNAMLRAHRLHQLAFDTYQEFEEAYIPKEQKYSFLQNPQTS LCFSESIPTPSNREETQQKSNLELLRISLLLIQSWLEPVQFLRSVFANSLVYGASDSNVYDLL KDLEE**K**IQTLMGRLEDGSPRTGQIFKQTYSKF|DDALLKNYGLLYCFRKDMDKV**E**TFLRI VQCRSVEGSCG

Where the "K" in bold type is the G120K substitution intended to destroy site 2 binding [2], and (for reference) "E" in bold type is E174. The pipe "|" indicates where we deleted the six-residue fragment "DTNSHN," which is blurred in PDB structures 1A22 and 1BP3, suggesting it is labile. This deletion makes the expressed protein more similar to that used in the modeling. The distance of this fragment from Site 1, plus its position within the longer, surface-exposed, mostly-random-coil residue stretch 129–154 means it is unlikely to affect receptor binding, as indeed is borne out by the agreement between WT and E174A affinities in our work vs. that of others.

In all cases, recombinant protein expression was carried out in XJB BL21(DE3) cells. rhGH was expressed by growing cells in LB until mid-exponential phase and induced with 1 mM IPTG (isopropyl-β-D-thiogalactopyranoside). Protein expression was completed with overnight growth at 16˚C for solubility, per [12]. Cells were harvested after two rounds of pelleting at $4,500 \times g$ for 10 min. Lysis of cell pellets and all subsequent purification were carried out at room temperature, unless otherwise noted. Cell pellets were resuspended in 10 mL of lysis buffer (50 mM Tris-HCl [pH 8.0], 100 mM NaCl, Roche Complete protease inhibitor) and lysed by a cell disruptor (Constant Systems Ltd, UK). Lysate clarification was achieved after 45 mins of centrifugation at 16,000 rpm with temperature set to 4˚C. Clarified lysate was filtered through a 0.45 μm membrane. Prior to loading the clarified lysate, the gravity column with 1ml of Ni-Sepharose resin was washed with 10 column volumes of ddH$_2$O and equilibrated with 10 column volumes of lysis buffer (10 mM Tris-HCl [pH 7.4], 500 mM NaCl, 5 mM imidazole). Lysate was applied directly to the capped gravity column and set on a shaker at 4˚C to gently mix for 40 minutes. After 40 minutes, lysate was eluted and a series of wash steps followed. The column was first washed with 5 column volumes of lysis buffer supplemented with 40 mM imidazole and then washed with 5 column volumes of lysis buffer supplemented with 10mM imidazole. The stepwise gradient was repeated a total of two times. rhGH was eluted into 1 mL fractions using 5 column volumes of 250 mM imidazole in lysis buffer. Protein

**Table 1. rhGH/ hGHR Binding energy predicted by HomologyScanner and SSIPe, and kinetics of binding measured by Surface Plasmon Resonance, ordered by $k_{off}$.** Green shading indicates $K_D$ comparable to that of WT. $\Delta\Delta G_{experimental}$ was computed based on the measured $K_D$ and T = 273K. In these cases, $k_{on}$ is also slower than WT, such that the only E174A has $K_D$ lower than that of WT. The Pearson correlation between $\Delta\Delta G_{homologyScanner}$ and $\Delta\Delta G_{experimental}$ was -0,18, and that between $\Delta\Delta G_{SSIPe}$ and $\Delta\Delta G_{experimental}$ was -0,40. This had no particular significance, with p-values of 0,4 and 0,07, respectively, by two-tailed t-test.

| Substitution | Class | HomologyScanner | SSIPe | Surface Plasmon Resonance | | | |
|---|---|---|---|---|---|---|---|
| | | ΔΔG [kcal/mol] | ΔΔG [kcal/mol] | $k_{on}$ [1000/Ms] | $k_{off}$ [0.001/s] | $K_D$ [nM] | ΔΔG [kcal/mol] |
| E174Y | polar | -0.88 | -0.331 | 8 | 0.03 | 4 | 1.63 |
| E174A | hydrophobic | -0.73 | 0.633 | 900 | 0.09 | 0.1 | -0.38 |
| H18F | hydrophobic | -0.56 | 0.804 | 300 | 0.09 | 0.3 | 0.22 |
| H18D | positive | -0.76 | 2.004 | 500 | 0.1 | 0.3 | 0.22 |
| E32A | hydrophobic | -0.86 | 0.195 | 800 | 0.2 | 0.3 | 0.22 |
| L52F | hydrophobic | -0.54 | 0.707 | 600 | 0.2 | 0.4 | 0.38 |
| Y28R | negative | -0.68 | 1.283 | 400 | 0.2 | 0.6 | 0.60 |
| WT | | | | 2000 | 0.3 | 0.2 | 0.00 |
| E65Y | polar | -0.65 | 0.478 | 700 | 0.3 | 0.4 | 0.38 |
| E174W | hydrophobic | -2.54 | -0.644 | 200 | 0.3 | 1 | 0.87 |
| E174R | negative | -0.75 | 0.84 | 7 | 0.3 | 40 | 2.87 |
| S62Y | polar | -1.25 | -0.17 | 600 | 0.4 | 0.5 | 0.50 |
| Q22H | polar | -0.74 | 0.56 | 50 | 0.4 | 8 | 2.00 |
| S62W | hydrophobic | -1.45 | -0.31 | 10 | 0.4 | 30 | 2.72 |
| M14E | positive | -0.91 | 0.00 | 40 | 0.5 | 10 | 2.12 |
| Q46R | negative | -0.84 | 0.44 | 10 | 0.6 | 40 | 2.87 |
| S51D | positive | -0.52 | 0.87 | 1000 | 0.7 | 0.6 | 0.60 |
| N63I | hydrophobic | -1.12 | 0.32 | 4000 | 1 | 0.3 | 0.22 |
| K41R | negative | -1.32 | 0.26 | 60 | 1 | 20 | 2.50 |
| R178Y | polar | -0.81 | -0.40 | 300 | 6 | 20 | 2.50 |
| R178F | hydrophobic | -1.22 | -0.33 | 300 | 8 | 30 | 2.72 |
| M14W | hydrophobic | -1.11 | 0.12 | 600 | 1000000 | 2 | 1.25 |
| E174D | positive | -1.38 | 0.46 | - | - | - | - |
| E32R | negative | -1.10 | 0.20 | - | - | - | - |
| Q46M | hydrophobic | -1.08 | -0.59 | - | - | - | - |
| E32H | polar | -0.88 | 0.20 | - | - | - | - |
| E65R | negative | -0.83 | 1.46 | - | - | - | - |
| S62K | negative | -0.71 | 1.25 | - | - | - | - |
| G190F | hydrophobic | -0.61 | 0.16 | - | - | - | - |
| H18S | polar | -0.55 | 1.85 | - | - | - | - |
| E65M | hydrophobic | -0.53 | 0.70 | - | - | - | - |
| T60F | hydrophobic | -0.52 | 0.17 | - | - | - | - |

purity was assessed by Commassie staining and fractions with the lowest impurities were pooled and concentrated to 250 μL using Amicon Ultra 4 mL Centrifugal Filters. A second round of purification was carried out via size exclusion chromatography, using the Agilent infinity 1220 HPLC system. The fractions containing protein were evaluated for purity using SDS gel electrophoresis and Coomassie staining. All fractions containing only rhGH were then pooled together and dialyzed at 4°C in 10 mM HEPES, 150 mM NaCl, 3 mM EDTA, 0.005% Tween 20, pH 7.4. After 24hrs of dialysis, protein was concentrated to 50 μL through the Amicon® Ultra 0.5 mL Centrifugal Filters. The concentration of final growth hormone purified using this method was assessed using a Bradford assay. The purified rhGH was then flash-frozen for storage.

## Interaction analysis using surface plasmon resonance (SPR) biosensor

The experiments were performed using a Biacore 3000 instrument (Cytiva, Uppsala, Sweden) at 25°C. The immobilization of hGHR was carried out by a standard amine coupling procedure on a CM5 biosensor chip (Cytiva, Uppsala, Sweden). hGHR was purchased from AbCam (ab180053) and diluted to 100 μg/ml in sodium acetate buffer, pH 4.5. The CM5 chip surface was activated by an injection of a 1:1 mixture of EDC and NHS for 7 min, at a flow rate of 10 μl/min. hGHR was injected over the activated surface at a flow rate of 2 μl/min until reaching an immobilization level of 1400–2700 RU. Then, the surface was deactivated by the injection of 1 M ethanolamine for 7 min. After immobilization, a concentration series of rhGH, ranging from 0.3 to 10 nM, was injected over the surface, at a flow rate 30 μl/min. An association phase was monitored for 60 sec and a dissociation phase for 240 sec. The surface was regenerated after each cycle by injecting 4.5 M MgCl2 for 2 min. The data was analyzed using Biaevaluation Software, v. 4.1 (Cytiva, Uppsala, Sweden). Sensorgrams were double-referenced by subtracting the signals from a reference surface and the average signals from two blank injections and fitted to a 1:1 Langmuir binding model [13].

## Circular Dichroism (CD) measurements

CD spectroscopy was used in order to identify the impact of each mutation on rhGH structure. 1 mL of purified rhGH samples was dialyzed against 2 L of 20 mM sodium phosphate buffer at pH 7.2 overnight to remove most of the tris buffer and salt used during purification via 2000-fold dilution. This dialyzed sample was dialyzed a second time overnight against and additional 2 L of fresh 20 mM sodium phosphate buffer at pH 7.2 to remove trace amounts of tris buffer and sodium chloride still present in the sample via a second 2000-fold dilution. This reduced the concentrations of both tris and sodium chloride in solution to 2.5 nM and 37.5 nM respectively and therefore minimized their impact on CD measurements.

The dialyzed sample was then loaded into a 10 mm length quartz cuvette. The sample was then loaded into the spectrophotometer (Chirascan). CD spectra between 240–190 nm were collected every two degrees centigrade as the sample was heated from 20–90°C, and then cooled back down to 20°C. The ratio of the measured ellipticity at 222 nm to the ellipticity at 208 nm was measured over a range of temperatures, for the WT and three variants.

# Results

## Circular Dichroism (CD) measurements

The CD measurements feature a clear isodichroic point, for WT, L52F, and E174A/Y. (Fig 2) The salient difference between the four variants is the relative depths of the troughs at 208 and 222 nm ($\theta_{222}/\theta_{208}$ ratio), which indicate an effect of E174Y and especially E174A on structure. The differences in $\theta_{222}/\theta_{208}$ ratio reduce with temperature, and disappear around 40–45°C. (Fig 3) L52F is not positioned to affect helix 1 & 4 structure and serves as a control for that purpose.

## Homologyscanner calculations

The homologyScanner results are given in Table 1. As can be seen, the computationally-selected mutations span all four physicochemical classes and are well spread out over site 1 (Fig 1). There was no particular correlation between $\Delta\Delta G_{homologyScanner}$ and $\Delta\Delta G_{experimental}$, see the Discussion. Other than E174A, no mutants had demonstrably higher affinity than WT. However we did identify a group of mutants which were comparable in affinity, some of which (interestingly) had a slower off-rate than WT.

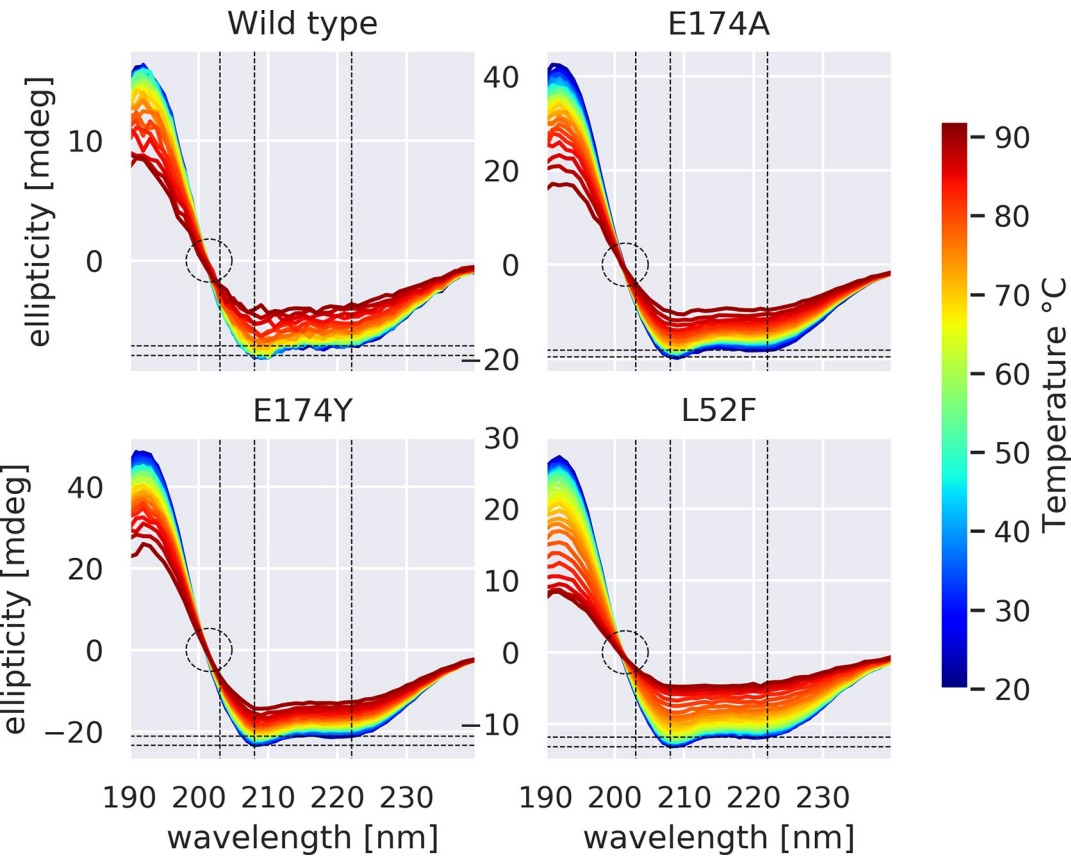

**Fig 2. Circular Dichroism spectra for WT and three rhGH variants, observed at temperatures from 20 to 92C.** Note the presence of clear isodichroic points at 201.5 nm (circled), indicating only α-helical and random-coil content (no β sheets), [14] consistent with crystallographic data (note this is clearly below 203 nm). The maximum at 190 nm and minima at 208 and 222 nm likewise indicate alpha helices. Higher ratio of ellipticity at 222 vs 208 nm (θ222/θ208) has been associated with coiled-coil content [15–17]. Dashed vertical lines indicate 203, 209, and 222nm. Dashed horizontal lines indicate the ellipticity at these two wavelengths and 20C –the smaller relative gap for E174A (compared to WT, E174Y, and L52F) is visually apparent. The isodichroic point (zero crossing) indicates the protein is two-state, coil and helix [18]; our isodichroic point is near the 201.5nm reported by [19], rather than the 203-205nm reported elsewhere [18].

## SSIPe calculations

Would another ΔΔG prediction method have given us better results? After the SPR experiment we retrospectively applied SSIPe, a newer method with good results on a benchmark dataset. SSIPe uses a sequence profile generated using PSI-BLAST, a structural profile generated with iAlign, and a physics-based energy function, EvoEF. [21] The results ($\Delta\Delta G_{SSIPe}$) are given in Table 1.

## Surface Plasmon Resonance (SPR) measurements

The SPR measurements produced clear $k_{off}$ and $k_{on}$ (given in Fig 4 and Table 1). The SPR sensograms are provided in S1 File. Our SPR experiments yielded an unexpected result. As mentioned, we produced 22 GH Variants: the wild-type (WT), positive control (E174A), and 20 novel mutants (including E174Y). In this text all comparisons of kinetic quantities ($K_D$, $k_{off}$, $k_{on}$, etc.) are with respect to WT. Recall that the dissociation rate is $K_D = \frac{k_{off}}{k_{on}}$.

E174A had a much lower $k_{off}$ but also a lower $k_{on}$–the $k_{off}$ won out leading to overall higher affinity (smaller $K_D$) compared to WT. 13 of the novel mutants had significantly higher $K_D$, and

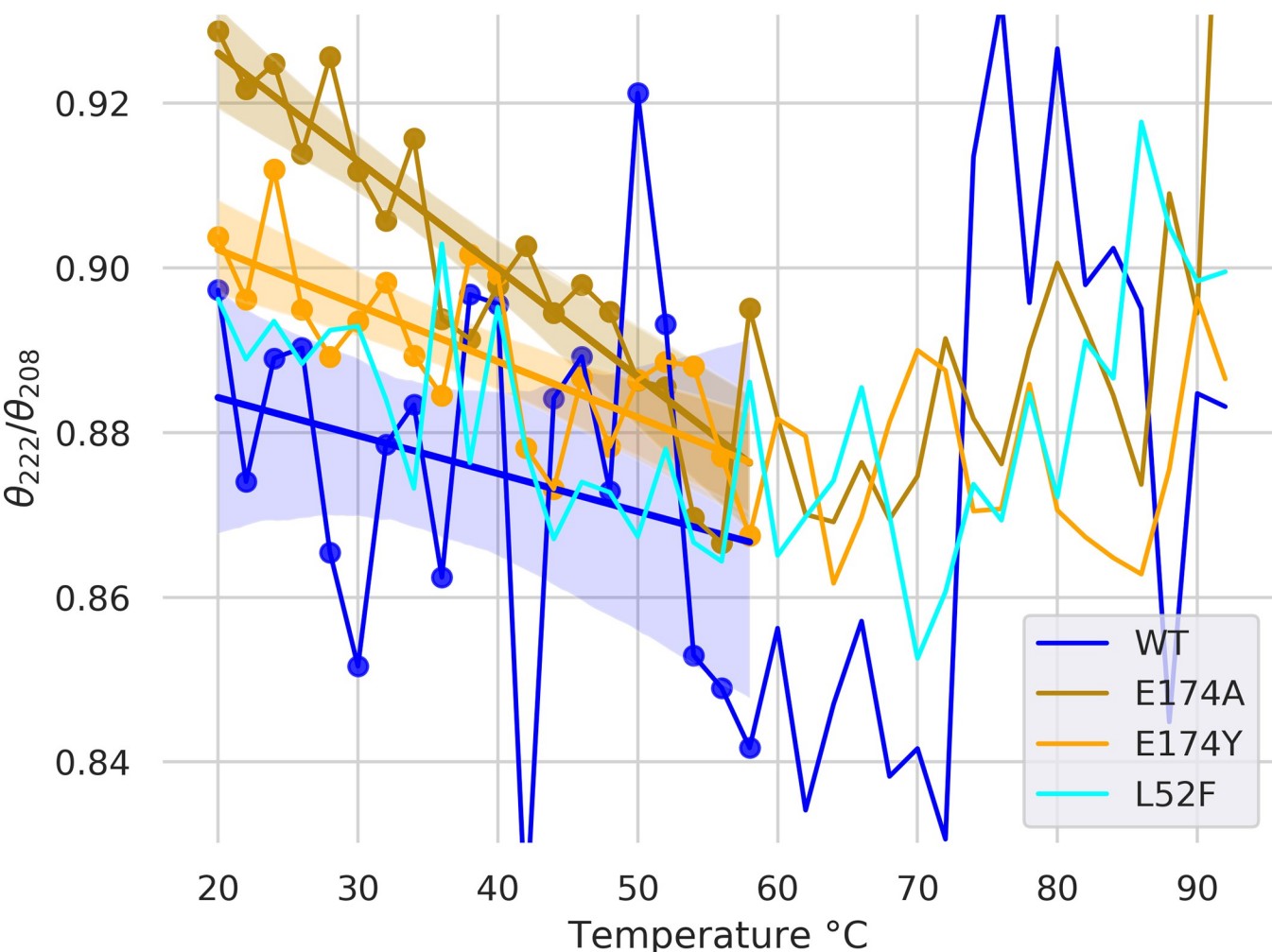

**Fig 3. Mutations at position E174 affect θ222/θ208.** We plotted θ222/θ208, for temperatures from 20 to 92C. Ratios greater than unity typically indicate coiled-coils. θ222/θ208 did not exceed unity for E174A. θ222/θ208 lower than 0.9 indicates helices in isolation [20], and indeed there is no clear coiled-coiling in experimental hGH structures (see PDB ID 1A22). Differences in ratio (comparing WT to E174A and E174Y) are greater at the lower end of the temperature range, presumably due to greater overall order. Linear regressions are shown for all variants (except for L52F, left out for clarity) over the temperature range of 20 to 58C (in which θ222/θ208 varies linearly). Also shown are 95% confidence intervals for the regressions (generated by the seaborn regplot function, in light shading). E174A has clearly higher θ222/θ208 than E174Y and WT. E174Y has somewhat higher θ222/θ208 than WT. L52F is a control, as it is not in a helix and not near the zinc-binding triad; its θ222/θ208 is indistinguishable from that of WT, and is less than that of E174A. E174A / E174Y increase θ222/θ208 ratio, while L52F has no clear effect.

in particular $k_{off}$ faster than WT. Six of the novel mutants (including E174Y) had slower $k_{off}$ and $k_{on}$, but unlike E174A in these six cases the slowdown in $k_{on}$ won out and the end result was a higher $K_D$. Interestingly, the highest-affinity mutants all decreased molecular mass (Fig 4).

## Discussion

The affinity increase induced by the E174A substitution (at 0.08nM, vs. 0.3nM for WT) was quite surprising at the time of its discovery [4] as alanine scanning usually disrupts rather than increasing binding. This position is in the zinc binding triad, and is highly conserved. In this work we confirm the earlier reported [4] increase in affinity of E174A, but now clarify that it is due to a decrease in $k_{off}$ which more than counters a decrease in $k_{on}$. The decrease in both quantities (albeit to varying degree), is shared by several other mutations. The mutations

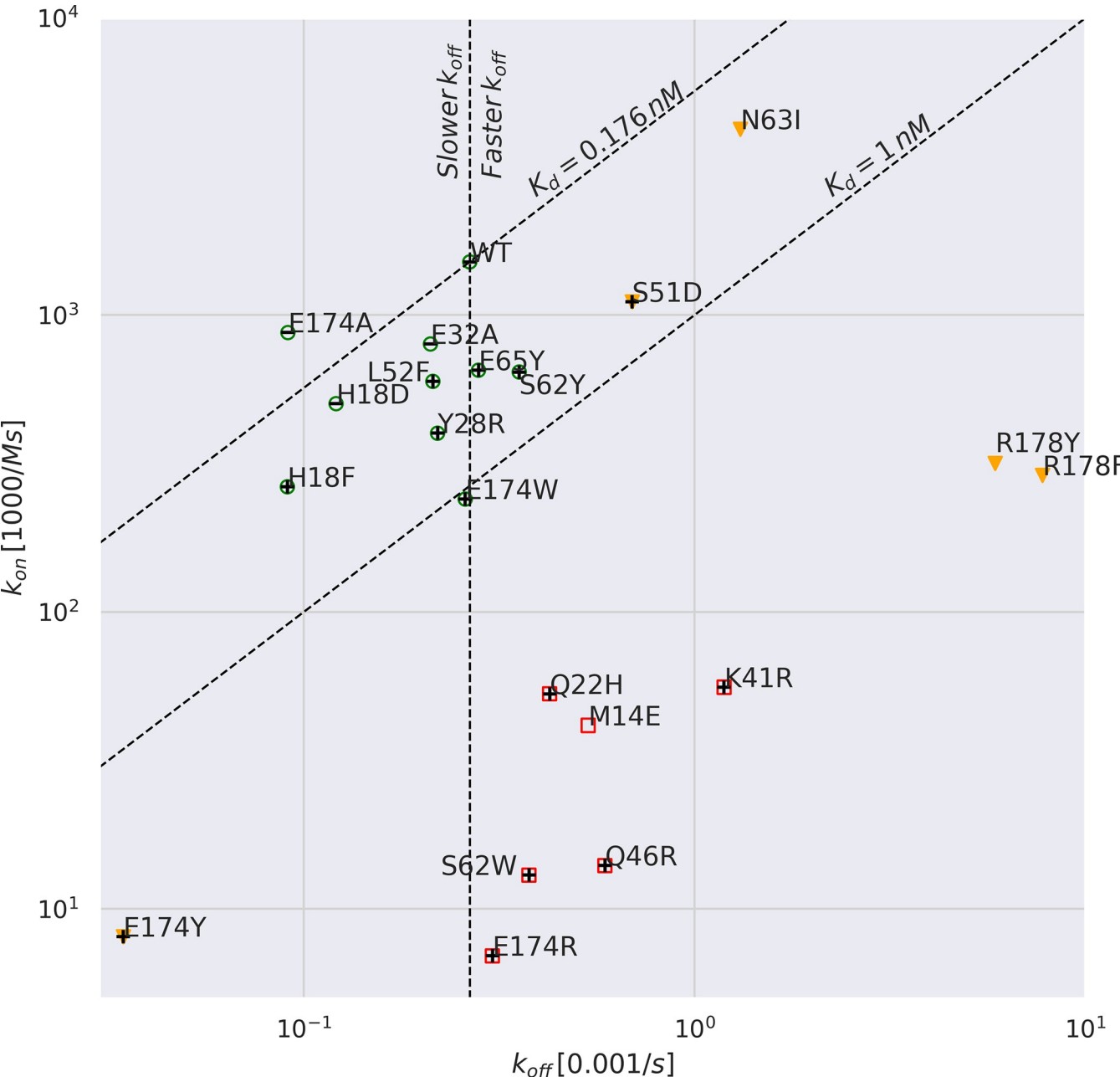

**Fig 4. $k_{on}$ vs $k_{off}$.** Dashed diagonals are iso-affinity contours (at WT and 1nM affinities). Vertical dashed line separates koff slower on left (blue circles) from faster than WT on right (red squares). M14W is off the scale (very fast koff, low affinity). We identified a cluster of higher- affinity mutants (green circles) of which many (including E174A) have slower koff than WT, but also slower kon. The slowest koff belong to substitutions at zinc binding triad positions H18 and E174 (we did not mutate the third position, H21). We also identified a group with moderately faster koff but much slower kon (red squares). Mutants outside these two groups are marked with yellow triangles. "+" markers superimposed on the above indicate an increase in molecular mass, from WT to mutant, while "-" markers indicate a decrease in molecular mass. Substitutions with small mass changes have neither of these markers. Note that the three mass-decreasing mutants (E174A, H18D, E32A) are among the four highest-affinity mutants. N63I, the fourth mutant, also has a (slight) decrease in mass.

E174A/Y increase $\theta_{222}/\theta_{208}$ ratio, at least in the absence of zinc. This suggests E174A and to a lesser extent E174Y may be inducing some amount of coiled-coiling.

The CD experiment was done with HPLC-purified rhGH variants–so no ions were present. E174 is part of the zinc-binding triad which by inspection would appear to stabilize the helix 1

–helix 4 interface. So for physiological zinc concentrations the $\theta_{222}/\theta_{208}$ ratio may change. Zinc-induced hGH homodimerization also competes with hGH-hGHR binding, and E174A, by affecting zinc binding, may reduce homodimerization to the benefit of hGH-hGHR binding (when zinc is present). In the absence of zinc, E174A could induce coiled-coiling. We speculate that it could do this by encouraging packing, though further experiments are needed to resolve this.

A separate matter is that of M14W, reported in [3], as a mutation that was overrepresented in a phage-display screen, and was included in their 852d multiple-substitution mutant (F10A, M14W, H18D, H21N, K41I, Y42H, L45W, Q46W, F54P, R64K, R167N, D171S, E174S, F176Y, I179T). M14W disappeared without clear explanation in the later B2036 mutant which formed the basis of Pegvisomant. F10A and F176Y also disappeared with little explanation, but did not have favorable HomologyScanner energy. Our results show that removing M14W was the correct decision on the part of [3].

The results also speak to the dearth of good published results for affinity maturation by computational selection of substitutions at a single position, compared to the many successes of combinatorial techniques like phage display [10]. As we have seen the former may change monomer properties, which the latter may compensate with substitutions at other positions. All potentials must balance the stabilizing effect of additional contacts, against the destabilizing effect of increased molecular volume. The latter can be due to entropy, desolvation, and sterics. The steric effect may be particularly important in our case since the mutations are mostly at alpha helical positions, which have little backbone flexibility. In [8], Dataset C, of 11 SKEMPI2 [22] mutations with $\Delta\Delta G_{experimental} < -0.7$ kcal/mol, only three increased mass, and included two positions in coils (1KAC, 1JTG) and one in a very short helix on the interface rim (2GYK). Many mass-increasing mutations were selected by our method, while only the mass-decreasing mutations yielded near-WT affinities, thus FoldX appears not to get this balance right in the context of helices in the PPI core [23]. In prior work we found near-WT affinities even with increases in molecular mass [24], but these were on the PPI rim [23], where there is more space available. Thus in future application of this method for that goal, one may consider preferring mass- or volume-decreasing substitutions for any positions in the core and on secondary-structural elements.

As noted in Table 1, there was no significant correlation between $\Delta\Delta G_{homologyScanner}$ and $\Delta\Delta G_{experimental}$. This could be expected because substitutions were *selected* for low $\Delta\Delta G_{homologyScanner}$ and so (aside from one -2.54 kcal/mol substitution) values range from -1.45 to -0.52 kcal/mol), over a small number of mutants. HomologyScanner has a reported [8] best-case Root Mean Square Error of 1.1 kcal/mol–about equal to the range of most of our data. SSIPe likewise showed no significant correlation.

## Conclusion

In summary, we have investigated the mechanism of the unexpected increase in hGHR-binding affinity of hGH substitution E174A. We found that it actually decreases the on-rate but overcompensates by decreasing the off-rate to an even greater extent. The CD results indicate the E174A mutation induces coiled-coiling.

All substitutions that yielded affinities comparable to wild type also decreased both on-rate and off-rate, though in cases other than E174A, the net effect was to slightly decrease affinity. Four of the highest-affinity substitutions decreased mass compared to WT, suggesting that there was not enough space or flexibility in these helices to accommodate any increases in side-chain size.

## Supporting information

**S1 File. Sensograms from surface plasmon resonance experiments.**
(XLSX)

**S2 File.**
(XLSX)

**S3 File.**
(XLSX)

**S4 File.**
(XLSX)

## Acknowledgments

The SciLifeLab Drug Discovery and Development Platform provided differential scanning fluorimetry equipment. Faruck Morcos helped us look for a coevoluationary signal in some rather limited sequence alignments.

## Author Contributions

**Conceptualization:** Samuel Coulbourn Flores.

**Formal analysis:** Samuel Coulbourn Flores.

**Funding acquisition:** Samuel Coulbourn Flores.

**Investigation:** Andrei Rajkovic, Joseph Erwin, Samuel Coulbourn Flores.

**Methodology:** Sandesh Kanchugal, Eldar Abdurakhmanov, Rebecca Howard, Sebastian Wärmländer, Astrid Gräslund, Helena Danielson, Samuel Coulbourn Flores.

**Software:** Samuel Coulbourn Flores.

**Supervision:** Sandesh Kanchugal, Astrid Gräslund, Samuel Coulbourn Flores.

**Visualization:** Samuel Coulbourn Flores.

**Writing – original draft:** Samuel Coulbourn Flores.

**Writing – review & editing:** Hugo A. Barrera Saldaña, Helena Danielson, Samuel Coulbourn Flores.

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
