## [Decision Letter · Decision Letter 0]

2 Jan 2023

PONE-D-22-33788Amino acid substitutions in human growth hormone affect secondary structure and receptor bindingPLOS ONE

Dear Dr. Samuel Coulbourn Flores,

Thank you for submitting your manuscript to PLOS ONE. After careful consideration, we feel that it has merit but does not fully meet PLOS ONE’s publication criteria as it currently stands. Therefore, we invite you to submit a revised version of the manuscript that addresses ALL the points raised by the two reviewers during the review process.

We look forward to receiving your revised manuscript.

Kind regards,

Sabato D'Auria

Academic Editor

PLOS ONE

“SF gratefully acknowledges a grant from the Swedish Foundation for International

Cooperation in Research and Higher Education (STINT). The Lars Hierta Memorial

Foundation provided on year of funding to AR. SF receives partial salary support from

the Swedish Research Council grant VR-M 2016-06301, the National Research School in

Medical Bioinformatics.

The SciLifeLab Drug Discovery and Development Platform provided differential

scanning fluorimetry equipment. Faruck Morcos helped us look for a coevolutionary

signal in some rather limited sequence alignments. The authors have no conflict of

interest.”

“We gratefully acknowledge support from the Swedish Foundation for International Cooperation in Research and Higher Education (STINT) to SF. The Lars Hierta Memorial Foundation provided salary support for AR. SF is funded by the Swedish Research Council grant VR-M 2016-06301, the National Research School in Medical Bioinformatics.”

4. We noted in your submission details that a portion of your manuscript may have been presented or published elsewhere. “An earlier version of this manuscript is on bioarxiv (doi: 10.1101/2021.12.16.473085)” Please clarify whether this publication was peer-reviewed and formally published. If this work was previously peer-reviewed and published, in the cover letter please provide the reason that this work does not constitute dual publication and should be included in the current manuscript.

5. Please amend either the abstract on the online submission form (via Edit Submission) or the abstract in the manuscript so that they are identical

6. Please include your tables as part of your main manuscript and remove the individual files. Please note that supplementary tables (should remain/ be uploaded) as separate "supporting information" files

Reviewers' comments:

Reviewer's Responses to Questions

**Comments to the Author**

1. Is the manuscript technically sound, and do the data support the conclusions?

Reviewer #1: Yes

Reviewer #2: No

2. Has the statistical analysis been performed appropriately and rigorously? 

Reviewer #1: I Don't Know

Reviewer #2: No

3. Have the authors made all data underlying the findings in their manuscript fully available?

Reviewer #1: Yes

Reviewer #2: No

4. Is the manuscript presented in an intelligible fashion and written in standard English?

Reviewer #1: Yes

Reviewer #2: No

5. Review Comments to the Author

Reviewer #1: The manuscript by Andrei Rajkovic et al. entitled “Amino acid substitutions in human growth hormone affect secondary structure and receptor binding” describes the interaction between human Growth Hormone (hGH) and hGH Receptor (hGHR) to select substitutions at single hGH positions to increase affinity of hGH against hGHR.

The authors conclude that the results point to a link between coiled-coilingstructure, zinc binding, and hGHR-binding affinity in hGH, and also suggest rules for choosing affinity- increasing substitutions.

I believe, I believe that reading the work is tiring. In my opinion the work should be much improved.

I admit this manuscript with major revision.

In particular:

- I would like to suggest to the authors to modify the introduction. It should contain an introductory part and then go into details.

- I would suggest trying to introduce mutations.

- The resolution of the figures is poor

- the discussion has no logical thread

- even if there are results, they are presented badly

- the work must be rewritten

The data is not well organized

Reviewer #2: The manuscript by Rajkovic et al. entitled “Amino acid substitutions in human growth hormone affect secondary structure and receptor binding” presents a study of the interaction between the human growth hormone and its receptor. Using a computational and experimental approach to find an action mechanism of 21 discovered mutations of hGH and their impact on the complex with hGHR. The mutations were modeled by FoldX. The mutations expressed and purified were experimentally studied with SPR and CD. About a previously known E174A mutation, the authors propose a mechanistic explanation for the increase in affinity, that involves part of a zinc-binding triad and packed in a coiled-coil interface. The aim of this work is interesting (and this reviewer appreciates it), although there are many inconsistencies between the theoretical and the experimental data that should be addressed in more detail. The paper is not well done in almost all parts, and the manuscript in its present form demands a deep revision before it can be published, so this reviewer suggests to reject the paper.

The paper should be modified in some parts.

Major issues:

1) The abstract needs to be revised according to the following suggestions.

2) The introduction section is too poor and unclear; nothing is reported on the aim of the paper.

3) The section Materials and Methods need to be reorganized and reordered. In this way is difficult to read and understand the methodologies used in the paper. About the CD, there are some inconsistencies regards the sample preparation (for example: by 2 cycles of dialysis is not possible to remove all the salt in the sample) and nothing is reported about the analysis of the collected data. Where is table 1? The SPR results are not well described.

4) As a consequence of the point before also the discussion section needs to be revised, this paragraph is really poor, no correlations were reported between the methodologies used, just hypothesis and open questions are reported. Where is table 1?

9) The conclusions section is expected by the reader, but in this paper is not present, mainly because there is nothing to conclude.

Minor issue:

1) Please renumbered the references, is difficult to consult them.

6. PLOS authors have the option to publish the peer review history of their article (what does this mean?). If published, this will include your full peer review and any attached files.

Reviewer #1: No

Reviewer #2: No

---

## [Author Response · Author response to Decision Letter 0]

31 Jan 2023

I provide full responses to the comments of Referees #1 and #2 in attached files.

---

## [Decision Letter · Decision Letter 1]

13 Feb 2023

PONE-D-22-33788R1Amino acid substitutions in human growth hormone affect secondary structure and receptor bindingPLOS ONE

Dear Dr. Samuel Coulbourn Flores,

Thank you for submitting your manuscript to PLOS ONE. After careful consideration, we feel that it has merit but does not fully meet PLOS ONE’s publication criteria as it currently stands. Therefore, we invite you to submit a revised version of the manuscript that addresses the minor points raised during the review process.

We look forward to receiving your revised manuscript.

Kind regards,

Sabato D'Auria

Academic Editor

PLOS ONE

Journal Requirements:

Reviewers' comments:

Reviewer's Responses to Questions

**Comments to the Author**

1. If the authors have adequately addressed your comments raised in a previous round of review and you feel that this manuscript is now acceptable for publication, you may indicate that here to bypass the “Comments to the Author” section, enter your conflict of interest statement in the “Confidential to Editor” section, and submit your "Accept" recommendation.

Reviewer #1: All comments have been addressed

Reviewer #2: All comments have been addressed

2. Is the manuscript technically sound, and do the data support the conclusions?

Reviewer #1: Yes

Reviewer #2: Yes

3. Has the statistical analysis been performed appropriately and rigorously? 

Reviewer #1: I Don't Know

Reviewer #2: Yes

4. Have the authors made all data underlying the findings in their manuscript fully available?

Reviewer #1: Yes

Reviewer #2: Yes

5. Is the manuscript presented in an intelligible fashion and written in standard English?

Reviewer #1: Yes

Reviewer #2: Yes

6. Review Comments to the Author

Reviewer #1: Dear Authors, the data has been better organized, even if the images of the graphs have low resolution.

Reviewer #2: (No Response)

7. PLOS authors have the option to publish the peer review history of their article (what does this mean?). If published, this will include your full peer review and any attached files.

Reviewer #1: No

Reviewer #2: No

---

## [Author Response · Author response to Decision Letter 1]

14 Feb 2023

We are pleased that the referees now find the work suitable for publication. We thank the Referees, Editor, and Editorial Staff for their work.

---

## [Editor Report · Decision Letter 2]

22 Feb 2023

Amino acid substitutions in human growth hormone affect secondary structure and receptor binding

PONE-D-22-33788R2

Dear Dr. Samuel Coulbourn Flores,

We’re pleased to inform you that your manuscript has been judged scientifically suitable for publication and will be formally accepted for publication once it meets all outstanding technical requirements.

Kind regards,

Sabato D'Auria

Academic Editor

PLOS ONE
---

## [Editor Report · Acceptance letter]

15 Mar 2023

PONE-D-22-33788R2 

Amino acid substitutions in human growth hormone affect secondary structure and receptor binding 

Dear Dr. Flores:

I'm pleased to inform you that your manuscript has been deemed suitable for publication in PLOS ONE. Congratulations! Your manuscript is now with our production department. 

Kind regards, 

on behalf of

Dr. Sabato D'Auria 

Academic Editor

PLOS ONE